# Implementing a Hand Gesture Recognition System Based on Range-Doppler Map

**DOI:** 10.3390/s22114260

**Published:** 2022-06-02

**Authors:** Yu-Chiao Jhaung, Yu-Ming Lin, Chiao Zha, Jenq-Shiou Leu, Mario Köppen

**Affiliations:** 1Department of Electronic and Computer Engineering, National Taiwan University of Science and Technology, Taipei 106335, Taiwan; d10902011@mail.ntust.edu.tw (Y.-C.J.); m10802129@mail.ntust.edu.tw (Y.-M.L.); m10902155@mail.ntust.edu.tw (C.Z.); 2Department of Computer Science and Systems Engineering (CSSE), Graduate School of Computer Science and Systems Engineering, Kyushu Institute of Technology, Iizuka-shi 820-8502, Japan; mkoeppen@ieee.org

**Keywords:** hand gesture recognition, FMCW radar sensor, range-Doppler map, deep learning, bidirectional long short-term memory

## Abstract

There have been several studies of hand gesture recognition for human–machine interfaces. In the early work, most solutions were vision-based and usually had privacy problems that make them unusable in some scenarios. To address the privacy issues, more and more research on non-vision-based hand gesture recognition techniques has been proposed. This paper proposes a dynamic hand gesture system based on 60 GHz FMCW radar that can be used for contactless device control. In this paper, we receive the radar signals of hand gestures and transform them into human-understandable domains such as range, velocity, and angle. With these signatures, we can customize our system to different scenarios. We proposed an end-to-end training deep learning model (neural network and long short-term memory), that extracts the transformed radar signals into features and classifies the extracted features into hand gesture labels. In our training data collecting effort, a camera is used only to support labeling hand gesture data. The accuracy of our model can reach 98%.

## 1. Introduction

There are many kinds of interfaces for human-computer interaction (HCI). Most commonly used interfaces are hand-manipulated [1,2], such as a mouse, keyboard, and touchscreen. During the COVID-19 pandemic, people desire to reduce the time spent touching their devices, which makes the solutions of the contactless HCI even more necessary. In this case, hand gestures will be practical in many fields.

Previously, studies related to hand gesture recognition were vision-based, usually using RGB cameras [3,4] or time-of-flight cameras [5,6,7], which have a high accuracy in recognizing whether an object is a hand or not. However, the hand gesture images captured from cameras usually contain personal or environmental information, which might cause privacy issues [8]. In addition, vision-based sensors are often sensitive to the environment, such as the intensity of light. Therefore, non-vision-based hand gesture recognition has been actively studied, such as using IR sensors, ultrasound, and radar [9,10,11,12,13,14,15,16,17,18,19]. These sensors have fewer privacy issues and consume low power. The data captured from these sensors is much simpler than images captured by cameras which makes it easier to be computed. However, the accuracy of determining whether an object is a hand or not might reduce.

In this paper, we will focus on hand gesture recognition with radar sensors. A radar sensor has more advantages in detecting moving objects; therefore, there are more studies about dynamic hand gesture recognition with radar than static hand gesture recognition. A recent study, the Soli project [13] from Google, shows the possibility of using radar in hand gesture recognition. The received signal from the radar is usually converted to data in the frequency domain such as range–Doppler maps (RDMs) or micro–Doppler images. These data in the frequency domain contain velocity and range information and are passed into the next stage of algorithms or models to predict hand gestures. In previous approaches, algorithms such as random forest trees or hidden Markov models were commonly used. With the advance in deep learning technology and computing power, neural networks are being used in an increasing number of applications to replace the traditional algorithms or machine learning models.

The position of the radar affects the effectiveness of these methods. The hand gesture set defined in each of the different studies also influences the chosen neural network architecture. In studies related to the Soli project [13,14], the radar was positioned underneath the user’s hand. The defined hand gesture set is based on micro hand gestures, which means it focuses on the variations between the user’s fingers and there is no directional hand gesture in this set [15]. Although these studies have good performance, these limitations make them not usable in some fields.

In this paper, we propose a dynamic hand gesture recognition system based on radar range–Doppler maps (RDMs). The hand gesture set we defined is based on the normal hand gestures that focus on the movement of the user’s palm, and our radar sensor is placed in front of the user’s hand. We propose a method to identify the specific direction of a hand gesture and locate the position of the hand relative to the radar sensor. We use a neural network (NN) for the feature extraction of the data as mentioned and long short-term memory (LSTM) model to classify the hand gestures.

In summary, we make the following contributions:An image-based radar data collection software;A trigger algorithm for data collection;A gesture recognition model architecture.

In Section 4.3, we show that the accuracy of predicting gestures using bidirectional long short-term memory (BiLSTM) is better than LSTM and has an accuracy of 98%. Additionally, on the data set collection, collecting more people but fewer data per person gives better training results than collecting fewer people but more data per person.

## 2. Background and Related Work

### 2.1. Vision-Based Hand Gesture Recognition

Most of the solutions for hand gesture recognition in early work are vision-based. The authors in [1] analyzed previous research related to hand gestures in HCI tasks, and then mentioned the data need to be processed in vision-based hand gesture recognition. Multiple machine learning algorithms for classification are also discussed in this research. In [3], the author reviewed the vision-based sensors and hand gesture databases used in vision-based hand gesture recognition.

### 2.2. Radar System

#### 2.2.1. FMCW Radar

Frequency modulated continuous wave (FMCW) radar is a radar system using frequency modulation techniques. In the FMCW radar system, the frequency of the transmitting wave increases linearly at a constant rate as shown in Figure 1. A short time of these signals is named a “fast time signal” or a “chirp”, and multiple collected chirps are called a “frame” or a “long time signal” [20].

#### 2.2.2. Hand Gesture Recognition with Radar System

Hand gestures also affect the methods of recognition. Gestures contain two movements of a human body which are the fingers and the palm. The main moving part of normal hand gestures is usually the palm. With only finger movement, these hand gestures are called micro hand gestures.

In [21], the authors proposed a method to detect the valid frames of hand gestures using micro–Doppler signatures of a CW radar. The accuracy of detecting valid frames is 96.88%. The authors in [22] used the micro–Doppler signature of a 77 GHz FMCW radar and trained a convolutional neural network (CNN) model for hand gesture recognition. This research shows the probability of predicting gestures from micro–Doppler signatures in different directions. Additionally, the classification accuracy with different incident angles was discussed. In [13], the author proposed a 60 GHz FMCW radar with two Tx and four Rx antennas. The RDMs were computed to the defined features and meta-features. The random forest classifier was used for classification and the result was filtered by a Bayesian filter. In [14], the same FMCW radar sensor in [13] was used. The hand gesture set in this research was a micro hand gesture set such as pinching index or finger sliding. The features of collected sequence RDMs were extracted by a CNN and classified by a recurrent neural network (RNN). In [23], four Rx antennas were located at four diagonal corners of the sensor board, and the Tx antenna was located at the center. The distance between the Rx and Tx antennas was 4.24 cm with a 24 GHz FMCW radar. The sequence RDMs from four Rx antennas were processed to obtain feature vectors named “projected RDM”, and passed to the input of LSTM for hand gesture recognition.

## 3. Proposed Method

### 3.1. System Description

The system flowchart is shown in Figure 2. The whole system can be divided into the following three steps:Preprocessing radar data;Data capturing;Classification.

These steps are discussed in the following sections.

### 3.2. Preprocessing Radar Data

The flowchart of preprocessing radar signals is shown in Figure 3. In the following section, we will describe how we pre-process the radar signal, the problems of the RDMs, and how we solved the problems.

#### 3.2.1. Processing Range–Doppler Maps

A typical way to process the radar signal is to transform chirps into micro–Doppler signatures, but these kinds of data do not have enough physical significance for hand gestures. Although we can still recognize hand gestures through this data with a machine learning algorithm, it is a challenge when we need more customization for our system. In this paper, we opt to use RDMs to recognize hand gestures since they have much information for us to make our system more flexible.

The signal data from radar antennas are collected as chirps, and chirps are stacked to a matrix. From these chirps, we can obtain the frequency spectrum by using a fast Fourier transform (FFT). The bins of the frequency spectrum refer to the range of targets from the radar. The range resolution ($d_{\mathrm{res}}$) of the radar sensor is given by
(1)$$d_{\mathrm{res}}=\frac{c}{2B}\;,$$
and the max range (*d_max_*) is given by
(2)$$d_{\max}=\frac{F_{s}{\;\times \;c\;\times \;T}_{c}}{2B}\;,$$
where c is the light speed (3 × 10^8^ m/s), B is the bandwidth of the radar sensor, *F_s_* is the analog-to-digital converter (ADC) sampling rate, and $T_{c}$ is the time separation between two chirps in a frame. Normally, there is a DC bin in the frequency spectrum that makes it unable to represent the true distance of objects. Thus, we perform mean removal on each chirp before the FFT process [24]. A comparison of doing mean removal or not is shown in Figure 4.

The range bins of each chirp are stored as the rows of a matrix. We then perform FFT across every column that resolves the targets in velocity. The velocity resolution ($V_{\mathrm{res}}$) of the radar is given by
(3)$$V_{\mathrm{res}}=\frac{\lambda }{{2T}_{f}}\;,$$
and the max velocity ($V_{\max}$) is given by
(4)$$V_{\max}=\frac{\lambda }{{4T}_{c}}\;,$$
where *λ* is the wavelength of the radar signal, *T_f_* is the frame time. We calculate the resolution and the max value of range and velocity through the above equations, and then the frequency bins can be mapped as range and velocity values; this process is called “range–Doppler transforming process”. In this way, a signal sequence received by the radar can be transformed into an RDM as shown in Figure 5. With RDMs, we can measure moving targets in front of our radar sensor such as a user’s hand.

#### 3.2.2. The Problems of Using Range–Doppler Map

Using only RDM as a feature to recognize hand gestures causes some problems. In this section, we describe these problems, and a way to solve them is described in Section 3.2.3.

##### The Position of Radar Sensor

Hand gestures have different patterns in the RDM when the radar is under the user’s hand, making these hand gestures easy to identify by using only RDM. In Figure 6, the radar sensor is placed underneath the user’s hand. The horizontal swipe has more speed variation but almost no range variation in its RDM. However, the vertical swipe has both. That makes the entire RDM sequence completely different. In Figure 7, the radar sensor is placed in front of the user’s hand. For horizontal and vertical swipes, they both have only speed variation but no range variation. That makes us not able to recognize these hand gestures by using only RDM.

##### No Direction in Range–Doppler Map

Another problem is that RDM does not contain direction information. It only shows range and speed values. That is, we are not able to recognize hand gestures with the same patterns but in different directions by only using RDM. Figure 8 shows the RDM of left and right swipes. Clearly, the patterns in the RDM of both two swipes are similar. Although we can measure the time difference of the same signal received from the transmitter antenna between multiple received channels to recognize the directions of hand gestures, this method needs a larger distance between antennas and is more difficult to deal with the RDMs when there is a requirement for system customization.

#### 3.2.3. Using Range–Angle Map

The two problems we mentioned above can be solved if we can obtain information about the object’s directions from radar data. This section introduces how we calculate the directional data and convert it to a more uncomplicated feature.

##### Setup Radar Antennas

To obtain the angle relative to the radar sensor of an object, the antennas of the radar need to be a uniform linear array (ULA). We need at least two antennas of a ULA for every direction we want. If we need both horizontal and vertical orientations, we need to set our antennas as an L shape, as in Figure 9.

##### Calculate Angle of Arrival

The angle of arrival (AoA) is a technique to calculate the distance and direction of a target by processing the received signal. The result of AoA can be determined by the measured phase difference in the 2D–FFT peak of each two different antennas in the antenna array. The schematic diagram is shown in Figure 10.

The measured phase difference (*ω*) can be expressed by Equation (5):(5)$$\omega =\frac{2\pi d\;\sin\theta }{\lambda }\;.$$

Here, *d* is the distance between two antennas, *λ* is the wavelength of the radar signal and *θ* is the AoA of the object. By inverting Equation (5), we can calculate the *θ* by the following:(6)$${\theta =\sin}^{-1}\left(\frac{\lambda \omega }{2\pi d}\right)\;\;.$$

Finally, we can obtain the AoA matrix in the same dimension as the RDM, which is named the range–angle map (RAM). An example of an RAM is shown in Figure 11. The *X*-axis in RAM refers to the angle, *Y*-axis refers to the range, and the value refers to the magnitude.

#### 3.2.4. Calculate Range–Angle Feature

After obtaining the RAM, we extract the range–angle feature (RAF) from RAM to reduce the complexity of the data. We pick one maximum value point from the RAM and use the range, velocity, and magnitude of that point as RAF. Then, we also normalize the range and velocity by dividing them by the max coordinate value of each axis. In our case, the size of RAM is 32 × 32, so we divided both range and velocity by 32. The feature extraction is defined as Equation (7):(7)S ∈ ℝ2, Sii=X, Y, XY=argmax(x, y)(RAMi(x, y)) ,∃(m, n) ∈ Si ,RAFi=[m/32n/32RAMi(m, n)]

In our case, we have an L shape antenna array which allows us to calculate three RDMs and the horizontal, vertical, and diagonal RAFs as shown in Figure 12.

### 3.3. Data Capturing

The simplest way to capture the time-sequenced radar frames is using a sliding window, but it is usually not efficient or flexible enough for most cases. Instead of using a sliding window, we design a trigger algorithm for data capturing which is shown in Figure 13. This algorithm determines a trigger flag that controls the capturing of radar data. Then the captured time-sequenced radar data is passed into the gesture recognition model.

By using the radar data from Section 3.2, we then choose one of the RDMs and perform the following image processing:Binarize the RDM with a threshold;Find contours in the binarized image;Get the max area contour.

A threshold for the contour area value ($\theta_{\mathrm{cnt}}$) is determined depending on the range value (*Y*-axis) of the max contour in RDM. Finally, a trigger flag is determined by $\theta_{\mathrm{cnt}}$ and the previous status of the trigger flag. We can also add different custom rules for different usage cases. For example, we might want the user’s hand gestures to start from the mid of the radar sensor so that we can add a rule to the RAFs data to make sure the start position of a hand gesture is in the middle of the sensor.

### 3.4. Gesture Recognition Model

We proposed a deep learning-based model to recognize the input radar data. The architecture is shown in Figure 14 and combines the three parts of our model.

#### 3.4.1. Feature Extractor

The general feature extractor architecture is CNN, which is widely used in tasks like image classification [25,26,27]. For micro hand gesture recognition, the position (refers to the range and velocity of the user’s hand) of the gesture pattern between the RDM frames is mostly static, but the pattern style changes. In this case, CNN is an effective feature extractor in which we do not care about the loss of position information due to convolutions. For normal hand gesture recognition, not only does the pattern between the RDM frames change but also the coordinate of that pattern. There are different coordinate changes on RDM for various hand gestures. The coordinate of the gesture data in a frame is also an important feature to recognize hand gestures, which means CNN might not be an effective feature extractor. Therefore, instead of using CNN, we use NN as our feature extractor. We design separate NN feature extractors for RDMs and RAFs from input frames. Since each RAF (in our case we have horizontal, vertical, and diagonal) represents different directional information, the NN feature extractor for each RAF is also distinct.

#### 3.4.2. RNN Model

In traditional machine learning, several algorithms are used to process time or space sequence data such as the Bayesian network or hidden Markov models (HMM). In deep learning, these tasks are usually performed by RNNs such as LSTM. In this paper, we use LSTM to recognize the feature extracted by NN. Two kinds of LSTM layer are used in our model, which is the normal LSTM layer and the BiLSTM layer. A bidirectional LSTM layer is a layer that not only reviews the sequence data from start to end as a normal LSTM layer does but also from end to start. Although a BiLSTM layer requires double parameters as the normal LSTM layer, it provides stronger robustness to our model. The comparison result of these two structures is discussed in the next chapter.

This section has reviewed the feature extractor model and the gesture recognition model. In the training stage, the two models are trained jointly. First, the frames of RDMs and RAFs obtained by the data capturing algorithm can be used to generate the gesture feature sequences by the feature extractor, and then we pass the gesture feature sequences to the LSTM layer and output them to a SoftMax layer for predicting gesture labels.

## 4. Experiments

In this section, we design several experiments to evaluate our proposed method. In Section 4.1, we first show the hardware setup and the environment of our experiment. We then describe the method of how we collect the hand gestures dataset and introduce the datasets we collected. In Section 4.3, we compare the model performance between two different model structures in the LSTM layer. In Section 4.4, we discuss the model efficiency when adding a few training data to train from an external dataset.

### 4.1. Data Collection

#### 4.1.1. Collecting and Labeling Data with an Image-Based Algorithm

The hardware setup for data collection is shown in Figure 15. The distance between the user’s hand and the radar sensor is about 40 to 75 cm. We set a camera on the top of the radar sensor.

To make it easier to collect hand gesture data, we developed a specific software. The architecture of the collecting software is shown in Figure 16. In this application, we used the data capturing trigger algorithm discussed in Section 3.3 to determine which frame is going to be collected. When capturing the radar hand gestures data, camera images are also collected, and it is important that the camera is only used for data collecting. We then designed an image-based hand gesture recognition algorithm to automatically label the collected data.

#### 4.1.2. Hand Gesture Dataset

The hand gesture set used in our experiment is illustrated in Figure 17. We design gestures that focus on the palm motion such as swiping, rotating, near, and away. The swipe gesture contains four different directions including swiping up, down, left, and right; the rotate gesture contains clockwise and counterclockwise rotating.

In dataset A, we have two users performing each hand gesture, and in dataset B, we have three users, for a total of five different users. The number of records is shown in Table 1. The frame lengths of the gesture data in these datasets are different, ranging from 5 to 49 (30 fps). Dataset A is used as a large dataset for training and dataset B is used as an external testing dataset. We also copied 30% of the data from dataset B and added it to dataset A as a new training dataset C.

### 4.2. Evaluation Metrics

In this paper, three evaluation metrics are used in our experiment, which are accuracy, recall, and precision.

#### 4.2.1. Confusion Matrix

A confusion matrix is a matrix that shows the prediction of the model and the true answer. A confusion matrix of two classes is shown in Table 2. In this paper, there are eight gestures in our experiment, so the confusion matrix of our model is a multi-class confusion matrix with a size of 8 × 8.

#### 4.2.2. Accuracy

Accuracy is a commonly used metric to evaluate the performance of a model. The accuracy is given by
(8)$$\mathrm{Accuracy}=\frac{\mathrm{TP}+\mathrm{TN}}{\mathrm{TP}+\mathrm{TN}+\mathrm{FP}+\mathrm{FN}}\;.$$

Since we have eight classes, the accuracy shown in the following sections is the average accuracy of the eight classes.

#### 4.2.3. Recall and Precision

Although we can preliminarily evaluate the model performance by using accuracy, accuracy is not a good metric according to the accuracy paradox. Recall and precision are better measures in most cases. The recall and precision are given by
(9)$$\mathrm{Recall}=\frac{\mathrm{TP}}{\mathrm{TP}+\mathrm{FN}}\;,$$
(10)$$\mathrm{Precision}=\frac{\mathrm{TP}}{\mathrm{TP}+\mathrm{FP}}\;.$$

The recall is used to evaluate how accurately a model predicts a positive class when the answer is positive such as malware detection. The precision is a metric to measure the misjudgment rate of the model which is used in cases like face recognition. The recall and precision shown in the following experiment are the average recall and average precision of eight classes.

### 4.3. Normal and Bidirectional LSTM Layer

From the discussion in Section 3.4, two different structures of the LSTM layer are tested in our research. The number of parameters of two LSTM layers is listed in Table 3 and Table 4 lists the hyperparameters of the model we used. As mentioned, the BiLSTM layer requires double parameters compared to the normal LSTM layer.

We used dataset A to train two different models. Figure 18 shows the result of the training. Comparing the two results, it can be seen that the accuracy of training is about 99% for both models. However, the BiLSTM layer has about 10% more accuracy in the validation of dataset B. Hence, although using a BiLSTM layer requires more training parameters, the improvement in accuracy makes it acceptable.

### 4.4. The Strategy of Data Collecting

From the result of previous experiments (Figure 18), it can be seen that there is an accuracy gap between the training stage and the validation of dataset B. The first reason is that our dataset is not varied enough, and the second is that the data distribution of the same person’s gestures looks quite the same, which makes the model validated by overlapping data.

To confirm our model has robustness, we need more data. That is, the strategy of data collecting is important. We then used dataset C mentioned in Section 4.1.2 to train our model and used the remaining 70% of dataset B as an external testing dataset; the result is shown in Figure 19 and Figure 20.

We had different strategies when collecting dataset A and dataset B. In dataset A, we collect fewer people’s hand gestures but more data per person; in dataset B, we collected more people’s hand gestures but fewer data per person. The result shows that there is no need to collect that much data per person to make the model recognize one person’s hand gesture. Additionally, the final accuracy of the validation of the external dataset is about 98%. Based on the result, we believe that our model can keep robustness with a more varied training dataset.

## 5. Conclusions

In this paper, we proposed a dynamic hand gesture recognition system based on an FMCW radar sensor. In the research, we calculate the RAF from the RDM of each radar antenna, which allows us to obtain information about the direction of hand gestures and locates the position of the user’s hand. The data capturing trigger algorithm captures the hand gesture data that need to be recognized. We then proposed an end-to-end trained model with an NN feature extractor and RNN. The accuracy of our model can reach 99.75% while training with our collected training dataset and reach 98.64% on an external testing dataset. From our experiments, we believe that our model can keep its robustness when facing a larger training dataset.

In future work, we will collect more users’ hand gesture data to make our training dataset more varied. In addition, we also plan to design a deep learning-based model for our data capturing trigger algorithm to make it more accurate in various scenarios. In addition, we plan to add more hand gestures to our hand gesture set to increase the number of available functions in our system.

## Figures and Tables

**Figure 1 sensors-22-04260-f001:**
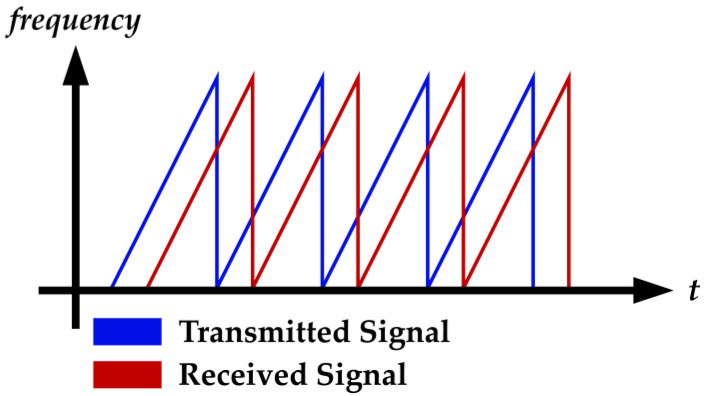
The chirps of the transmitted and received signals.

**Figure 2 sensors-22-04260-f002:**
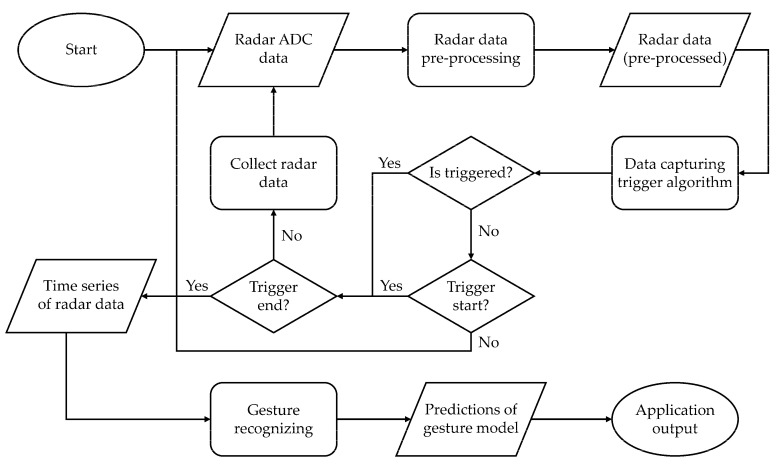
System flowchart.

**Figure 3 sensors-22-04260-f003:**
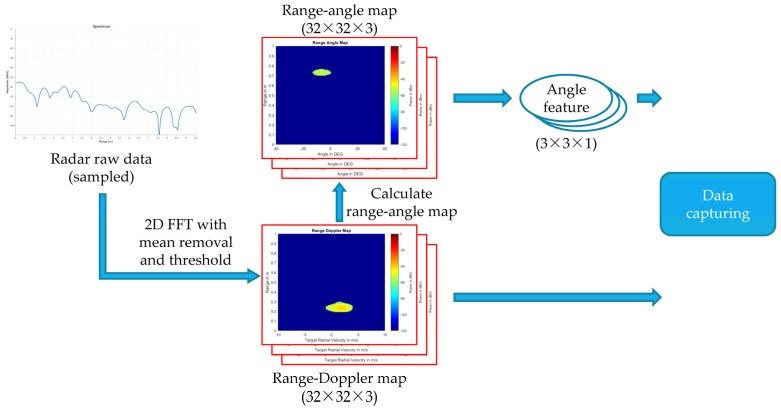
Flowchart of preprocessing radar signals.

**Figure 4 sensors-22-04260-f004:**
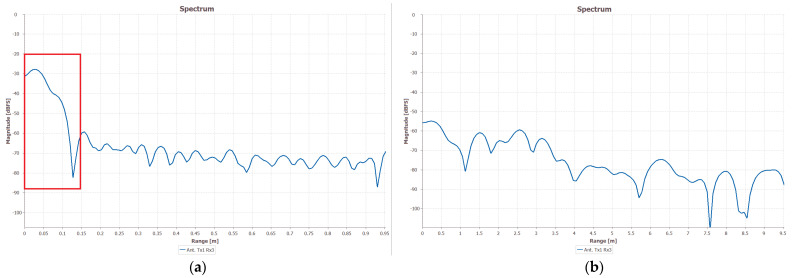
Range profile with mean removal. (**a**) Without mean removal. The red block shows the DC bin in the spectrum. (**b**) With mean removal. The DC component is reduced.

**Figure 5 sensors-22-04260-f005:**
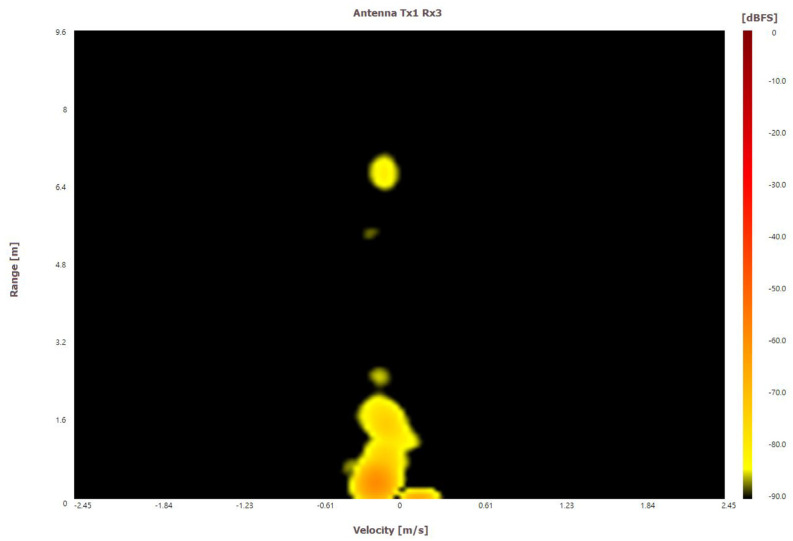
A range–Doppler map (RDM). The *X*-axis represents the velocity, and the *Y*-axis represents the range.

**Figure 6 sensors-22-04260-f006:**
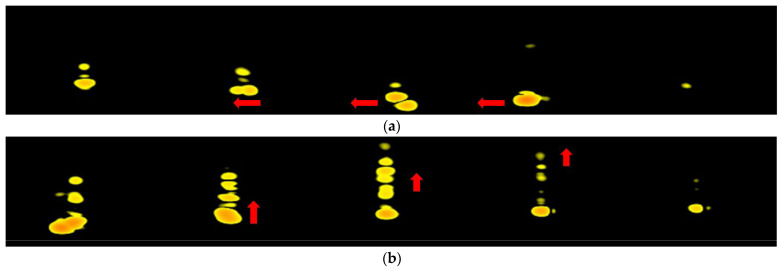
The RDM of horizontal and vertical swipe. (Radar is underneath the user’s hand.) (**a**) Horizontal swipe. The arrow shows the changes in the velocity between frames. (**b**) Vertical swipe. The arrow shows the changes in the range between frames.

**Figure 7 sensors-22-04260-f007:**
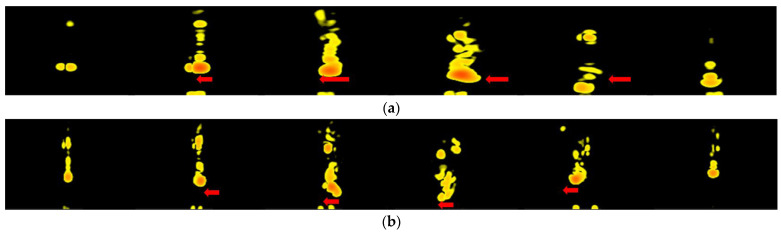
The RDM of horizontal and vertical swipe. (Radar is in front of the user’s hand.) (**a**) Horizontal swipe. The arrow shows the changes in the velocity between frames. (**b**) Vertical swipe. The arrow shows the changes in the range between frames.

**Figure 8 sensors-22-04260-f008:**
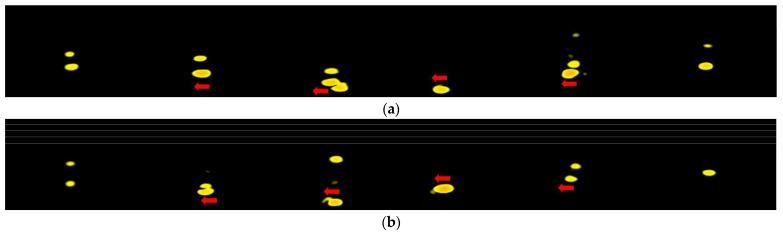
The RDM of left and right swipes. (**a**) Left swipe. The arrow shows the changes in the velocity between frames. (**b**) Right swipe. The arrow shows the changes in the velocity between frames.

**Figure 9 sensors-22-04260-f009:**
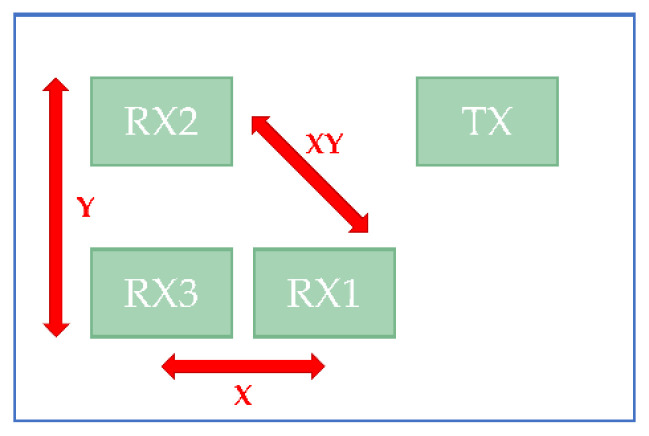
The minimum requirement of antennas for horizontal and vertical directions.

**Figure 10 sensors-22-04260-f010:**
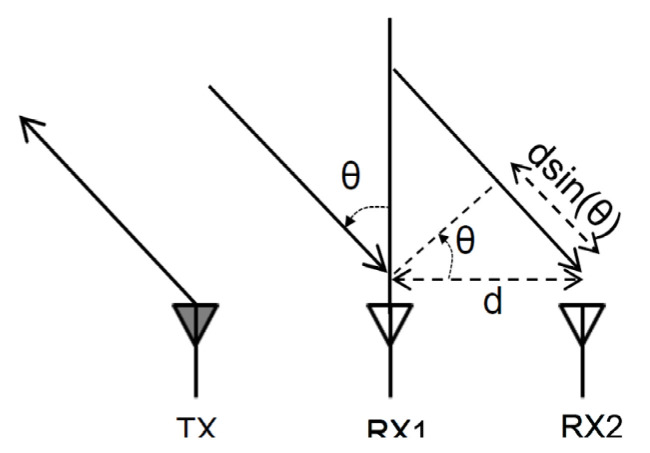
The schematic diagram of measuring angle of arrival (AoA).

**Figure 11 sensors-22-04260-f011:**
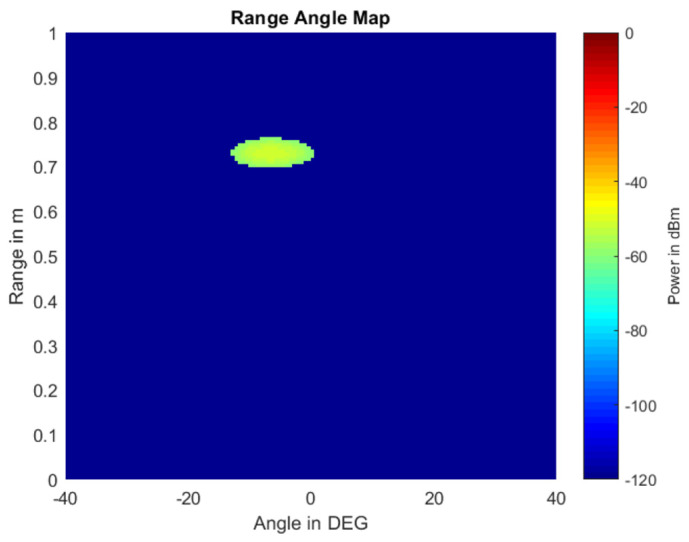
Range–angle map.

**Figure 12 sensors-22-04260-f012:**
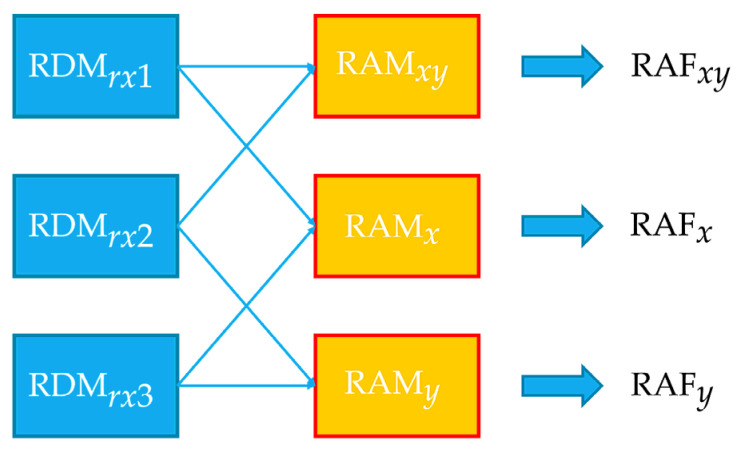
Diagram of calculating RAFs.

**Figure 13 sensors-22-04260-f013:**
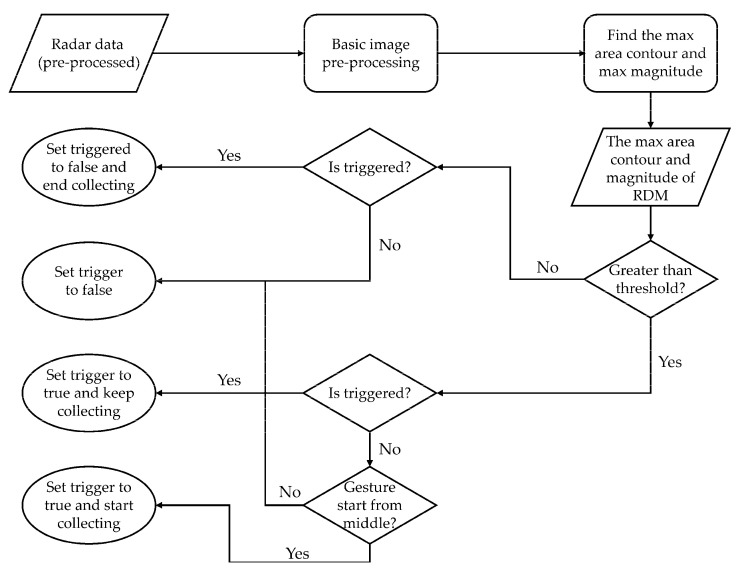
Flowchart of data capturing trigger algorithm.

**Figure 14 sensors-22-04260-f014:**
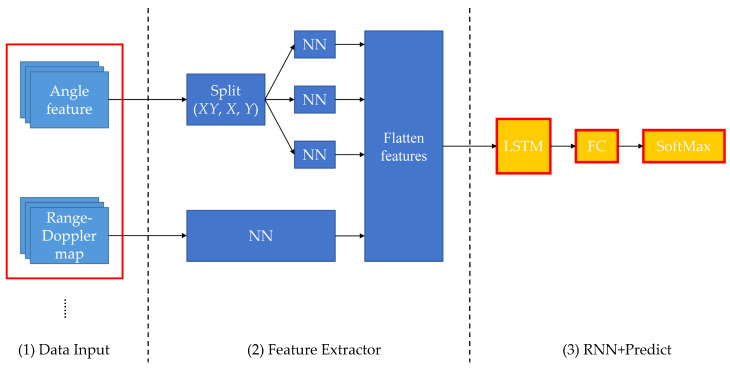
Architecture of hand gesture recognition model.

**Figure 15 sensors-22-04260-f015:**
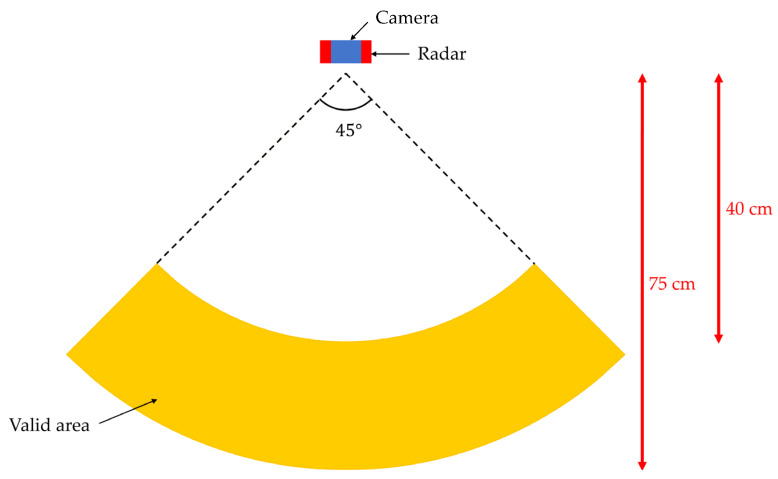
The environment of our experiment.

**Figure 16 sensors-22-04260-f016:**
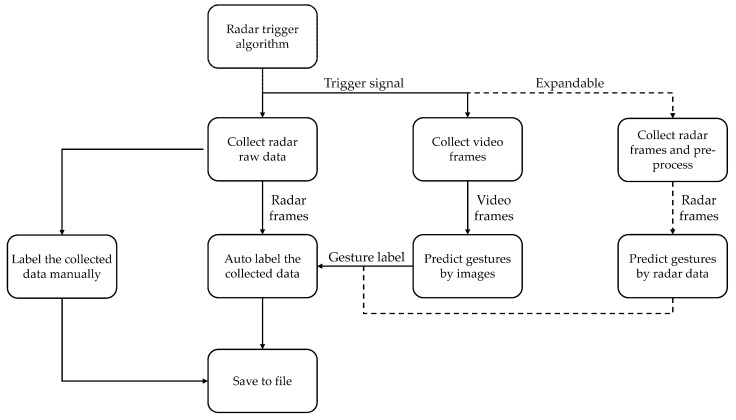
The architecture of data collecting software.

**Figure 17 sensors-22-04260-f017:**
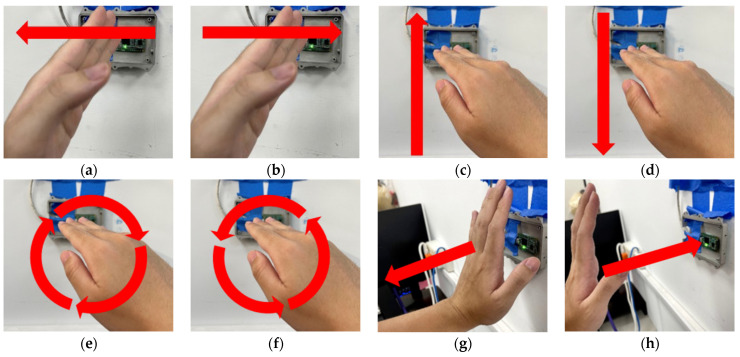
The hand gestures set used in our experiment. (**a**) Swipe left. (**b**) Swipe right. (**c**) Swipe top. (**d**) Swipe down. (**e**) Clockwise. (**f**) Counter-clockwise. (**g**) Near. (**h**) Away.

**Figure 18 sensors-22-04260-f018:**
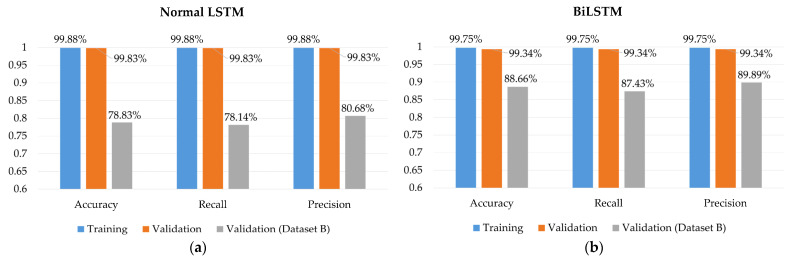
Model performance on two different LSTM layers. (**a**) Normal LSTM layer. (**b**) BiLSTM layer.

**Figure 19 sensors-22-04260-f019:**
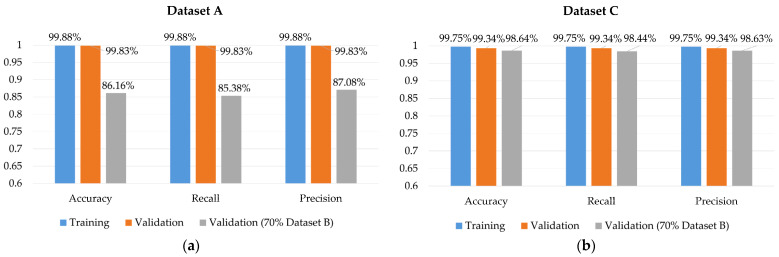
Model performance on two different datasets. (**a**) Training with dataset A. (**b**) Training with dataset C.

**Figure 20 sensors-22-04260-f020:**
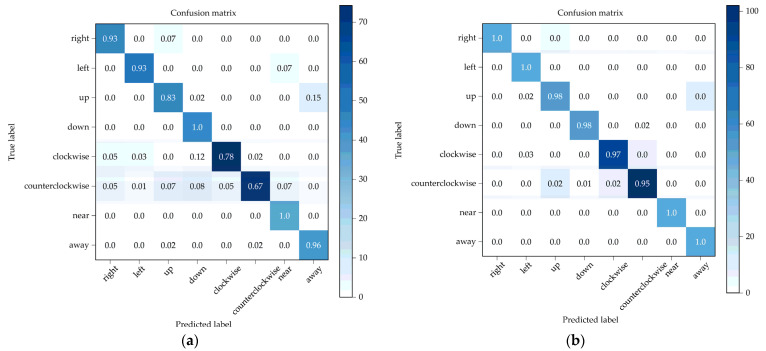
Confusion matrix of two different datasets. (**a**) Dataset A. (**b**) Dataset C.

**Table 1 sensors-22-04260-t001:** Numbers of datasets.

Dataset	Users	Horizontal Swipe	Vertical Swipe	Rotating	Near/Away	Total Records
Dataset A	2	Swipe left: 289	Swipe up: 306	Clockwise: 272	Near: 120	2012
		Swipe right: 300	Swipe down: 300	Counter-clockwise: 309	Away: 116	
Dataset B	3	Swipe left: 82	Swipe up: 82	Clockwise: 139	Near: 59	732
		Swipe right: 85	Swipe down: 82	Counter-clockwise: 144	Away: 59	

**Table 2 sensors-22-04260-t002:** Confusion matrix.

		Predicted Class	
		Positive	Negative
Actual class	Positive	TP	FN
	Negative	FP	TN

**Table 3 sensors-22-04260-t003:** The numbers of parameters of two different LSTM layers.

Model Structure	Parameters of Feature Extractor	Parameters of LSTM Layer	Total Parameters
Normal LSTM	5164	8256	13,556
BiLSTM	5164	16,512	21,940

**Table 4 sensors-22-04260-t004:** The hyperparameters of the model.

Parameter	Values of the Parameter
Batch size	32
Number of hidden units of LSTM	16
Number of hidden units of NN	8, 16, 32
Pool size of Max pooling	(2, 2)
Number of iterations (epochs)	150
Steps per epoch	49
Validation steps	49
Optimizer	Adam
Learning rate	0.001

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
