# Peer review of "Implementing a Hand Gesture Recognition System Based on Range-Doppler Map"

_sensors, 2022, doi:10.3390/s22114260_

Round 1

Reviewer 1 Report

The authors proposed a method for the recognition of hand gestures based on range-Doppler map. In my opinion the paper may be reconsidered for publication after the authors address the following issues.

  1. What are the main contributions of this work? They should be specified in Section 1 or 2.
  2. There are no "Yes" and "No" after the block "Trigger end" in the flowchart from Fig. 2.
  3. Are the parallelograms in the flowchart from Fig. 2 supposed to be data and rectangles supposed to be methods? If so, the caption of the first block (after the START block) should be "Radar ADC data" instead of "Get radar ADC data". There are also inconsistencies of shapes of the blocks "Gesture model" and "Prediction".
  4. The authors should write about the parameter values of LSTM and NN used in their experiments, such as: mini batch size, number of hidden units, number of iterations (epochs), optimizer, initial learning rate.
  5. "We also sampled 30 % of the data from dataset B and added it to dataset A as a new training dataset C.". Does that mean there are some identical samples in dataset A and B? Are these samples copied or transferred?
  6. There are three subjects in dataset A and two subjects in dataset B. Are these same people or different people? It should be specified in the text.
  7. In Section 4.2.1. the authors write about confusion matrix but no confusion matrix is presented. Please provide confusion tables from the experiments with names of gesture classes in rows and columns.

Language

  • The following words and sentences require correction:
  • In references of Sections, there should be space between "Sec." and numbers. E.g., "Sec. 4.1" instead of "Sec.4.1".
  • "interfaces for human-computer interaction (HCI)" - lack of period at the end of the sentence.
  • "In this section, we describe these problems and how we solve them in Sec.3.2.3." - I suggest writing "In this section, we describe these problems, and a way to solve them is described in Sec. 3.2.3." or something like that.
  • "That makes the entire RDM sequence is completely different." - it should be "That makes the entire RDM sequence completely different." (without "is").
  • "That makes us not able to recognize these hand gestures by only using RDM." - I suggest writing "by using only RDM".
  • "The architecture is shown in Figure 14, combines the three parts of our model." - I think it should be "in Figure 14, and combines the three parts".
  • "In this case, CNN is an effective feature extractor in that we don’t care about the loss" - I suggest writing "in which we don't care".
  • "to recognize hand gestures which means CNN might not be" - it should be "hand gestures, which means" (with comma).
  • "data capture by the data capturing trigger algorithm" - I think it should be "captured".
  • "(refers to user hand’s range and velocity)" – I suggest writing "(refers to the range and velocity of the user's hand)".
  • "camera images are also collected, it is important that the camera" - I think it should be "camera images are also collected, and it is important that the camera".
  • "hand gesture recognition algorithm for automatically labeling the data we collected." - I suggest writing "for automatic labeling of the collected data" or "to automatically label the collected data".
  • "we design several experiments to confirm our proposed method" - I suggest writing "to evaluate our proposed method".
  • "We design gestures that focuses on the palm motion" - It should be "focus".
  • in Fig. 17 - "(g) anear" - I think it should be "near".
  • "to evaluate how accurate a model predicts a positive class" - it should be "how accurately".

Author Response

Thank you for reviewing.

Reviewer 2 Report

In this paper the authors propose a methodology for hand gesture recognition based on range-Doppler maps and deep learning, and they prove to achieve an excellent accuracy of 98%. The considered topic is interesting and timely, however, I have the following main concerns:

  1. The Related work section is not sufficient, more recent paper should be included, such as
  • Pramudita, Aloysius Adya. "Contactless hand gesture sensor based on array of CW radar for human to machine interface." IEEE Sensors Journal 21.13 (2021): 15196-15208.
  • Senigagliesi, Linda, Gianluca Ciattaglia, and Ennio Gambi. "Contactless walking recognition based on MMwave radar." 2020 IEEE Symposium on Computers and Communications (ISCC). IEEE, 2020.
  • Magrofuoco, Nathan, et al. "Eliciting contact-based and contactless gestures with radar-based sensors." IEEE Access 7 (2019): 176982-176997.
  1. Moreover, the authors should better highlight the novelty of their work with respect to the state-of-the-art; in this regard, it would be useful to include comparison with other radars of the same type (FMCW), for example working at other frequencies.
  2. The difference between RDM and RDI is not clear, sometimes they seem to be used interchangeably and sometimes they are not.
  3. I would like to see the confusion matrix with the different gestures; are the users the same for both dataset A and B?
  4. The quality of the figures should be improved; also, the flowcharts are not very clear: in Figure 2 I think some yes/no pairs are missing, and I cannot understand the usefulness of the one in Figure 16.
  5. The use of English should be improved, there are numerous typos.

Author Response

Thank you for reviewing.

Round 2

Reviewer 1 Report

The authors addressed all of my concerns. The paper is better now. However, I still have two comments:

1. Table with hyperparameters from the answer of the fourth comment should be written in the paper text.

2. The caption of Fig. 20 -  "Confusion matrix of two different various datasets. (a) Dataset A. (b) Dataset C.". It should be "two different" or "two various". There is no need to write "two different various".

Author Response

It’s our pleasure to have the opportunity to get a chance to revise our paper (sensors-1722382). Thanks for your valuable comments. Your detailed and constructive comments have greatly helped us improve the quality of this paper. Following your comments, we have revised our paper accordingly and highlighted the improved parts in the revised manuscript. Our responses to the editor's and reviewers’ comments are illustrated below.

1. Table with hyperparameters from the answer of the fourth comment should be written in the paper text.

Response:

Thanks for the reviewer's comment, we have added the table with hyperparameters at Sec. 4.3 as Table 4.

2. The caption of Fig. 20 -  "Confusion matrix of two different various datasets. (a) Dataset A. (b) Dataset C.". It should be "two different" or "two various". There is no need to write "two different various".

Response:

We appreciate the reviewer for pointing out our mistake. We have modified the caption from "... two different various..." to "two different".

Reviewer 2 Report

The authors have addressed all my comments and revised the paper accordingly. 

Author Response

Thank you for reviewing.

Best regards, 

Yu-Chiao Jhaung , Yu-Ming Lin , Chiao Zha , Jenq-Shiou Leu , Mario Köppen